# Kangaroo Mother Care in Vietnam: A National Survey of a Middle-Income Country

**DOI:** 10.3390/children9111667

**Published:** 2022-10-31

**Authors:** Francesco Cavallin, Daniele Trevisanuto, Tran Viet Tiep, Nguyen Thi Ngoc Diep, Vuong Thi Hao, Doan Thi Ngan, Nguyen Thi Thuy, Nguyen Thi Xuan Hoi, Luciano Moccia

**Affiliations:** 1Independent Researcher, 36020 Solagna, Italy; 2Department of Woman and Child Health, University of Padova, 35128 Padova, Italy; 3Vietnam-Sweden Uong Bi General Hospital, Uong Bi 02306, Vietnam; 4Pediatrics Department, Vietnam-Sweden Uong Bi General Hospital, Uong Bi 02306, Vietnam; 5Newborn Department, Vietnam-Sweden Uong Bi General Hospital, Uong Bi 02306, Vietnam; 6Research Office, Training and Direction of Health Activities Center, Vietnam-Sweden Uong Bi General Hospital, Uong Bi 02306, Vietnam; 7Day One Health South East Asia, Hanoi 131000, Vietnam; 8Day One Health, Redding, CA 96001, USA

**Keywords:** breastfeeding, kangaroo mother care, middle-income country, survey

## Abstract

Background: Kangaroo mother care (KMC) is a low-cost intervention that is indicated to be a highly effective practice for which adoption and implementation are lacking. We investigated the current provision of KMC in Vietnam and explored differences among levels of healthcare facility. Methods: A survey form was sent to 187 hospitals in Vietnam, representing the three levels (central, provincial and district) of public hospital-based maternity services. Results: Overall response rate was 74% (138/187 hospitals). Routine KMC implementation was estimated in 49% of the hospitals. Where KMC was implemented or was being introduced, half of the hospitals had a written protocol and a KMC-dedicated room, and held educational courses on KMC. KMC was mainly performed by the mother. Skin-to-skin contact was mostly performed for <12 h/day (55%), exclusive breastfeeding at discharge was very frequent (89%) and early discharge was considered in half of the hospitals (54%), while follow-up was not performed in 29% of the hospitals. Participants considered follow-up after discharge as the main barrier to KMC implementation, and indicated education (of both parents and health caregivers) and environment upgrades (KMC-dedicated room and equipment) as the most important facilitators. Conclusions: Our survey estimated a limited implementation of KMC in Vietnamese maternity hospitals, with marked variations across the different levels of maternity services. Areas of improvements include increasing the duration of skin-to-skin contact, arranging dedicated spaces for KMC, involving the relatives (especially at district level), extending the availability of a written protocol, improving the eligibility process, and implementing early discharge and follow-up monitoring.

## 1. Introduction

Kangaroo mother care (KMC) is a low-cost intervention that includes prolonged skin-to-skin contact between mother and newborn, exclusive breastfeeding, early discharge from the health facility, and close follow-up at home [1,2]. KMC may prevent many complications associated with preterm birth and may also provide benefits to full-term newborns [3,4,5,6,7]. Recent systematic reviews outlined that KMC is beneficial on temperature control, growth rate, infection rate and neurodevelopmental outcome [1,3].

The World Health Organization (WHO) guidelines strongly support KMC [2,8,9], which has also been identified as a high-priority intervention for preterm newborns [10]. However, country-level adoption and implementation of KMC has been limited [11]. The slow uptake of KMC is indicated as an example of highly effective intervention for which adoption and implementation have lagged [11]. In 2013, a group of newborn health stakeholders discussed barriers to implementation and research priorities for KMC uptake [12]. The consensus meeting indicated several bottlenecks for KMC uptake, including leadership/governance issues, financial resources, health staff capacity and training, and lack of KMC coverage data [12]. Indeed, Darmstadt et al. underlined that data on KMC coverage are often unavailable because they are not integrated into country-level health information systems or periodic surveys [13]. Therefore, implementation of KMC in high-, middle- and low-resource settings remains a goal for the health care system. Gathering information on actual KMC care delivery is crucial as the background for understanding the barriers to application of KMC in practice, and for planning interventions to improve KMC implementation [12].

The purpose of this survey was to describe the current provision of KMC in Vietnam and to investigate differences among levels of healthcare provision.

## 2. Methods

### 2.1. Design

A structured, cross-sectional survey on KMC practice was conducted among public hospital-based maternity services in Vietnam, using an approach that was previously implemented for similar research questions on neonatal resuscitation [14,15]. In this country, the health system delivers care through four overlying administrative levels (central, provincial, district and community), where the community level does not provide hospital-based maternity services [14]. This study used hospital-level aggregate data, information on procedures and activities related to KMC, and opinions about barriers to and facilitators of the implementation of KMC. The study did not include any patient personal data or identifiable patient information, and did not involve patients or parents at any stage.

### 2.2. Participants

One Day Health [16], a non-government organization engaged in neonatal care in Vietnam, created a listing of all maternity hospitals in Vietnam using available data and documentation [14,15]. The sampling frame for the study included the 610 maternity hospitals with 500 or more births per year, representing 97% of maternity hospital births [14]. This list was used to create two samples: the first included the census of all 6 central hospitals and 72 provincial hospitals, while the second included a 20% sample survey of 532 district hospitals. These were randomly selected from each Vietnamese administrative region: each region provided 20% of the district hospitals in its catchment, using a stratified random sampling with proportional allocation approach [14,15]. Overall, 187 hospitals (6 central hospitals, 72 provincial hospitals, 109 district hospitals) were contacted to participate in the survey.

### 2.3. Procedures

A structured 37-item questionnaire and an enclosed introductory letter were sent by email to the directors of the obstetrical/neonatal wards of the participating hospitals. To increase the response rate, we sent a reminder to the non-responders (every two weeks for a maximum of three times); if unsuccessful, we tried a phone contact and sent a new email with the invitation. Participation was voluntary. The survey was conducted between January and September 2020.

The questionnaire included different sections about hospital information, organizational and practical aspects of KMC in the participating hospital, and opinions on barriers to and facilitators of the implementation of KMC as perceived by the responders. The content on organizational and practical aspects of KMC was based on dedicated KMC literature [1,8,9,11], while barriers and facilitators were chosen according to the 2013 consensus meeting [12]. The questionnaire included multiple-choice and fill-in items. The English-language questionnaire was revised for the Vietnamese context by the researchers from Vietnam-Sweden Uong Bi General Hospital (Quang Ninh Province, Vietam). The English-language questionnaire was translated into Vietnamese and pre-tested before the diffusion. Participants filled in the questionnaire and sent it back to the study logistics office of One Day Health in Hanoi (Vietnam).

Data were extracted from each questionnaire and recorded in a dedicated data sheet by the staff in the study logistics office. The staff was also available for providing information and clarification to participants.

### 2.4. Statistical Analysis

Data analysis was performed according to the sampling frame (as described before). Estimates at central level were the percentages calculated among responders, as the study included the census of all central hospitals (with 100% response rate). As the response rate was <100% at the provincial and district levels, data were inflated by the inverse of the response rate to avoid systematic bias. Provincial estimates (with 95% confidence intervals) were the weighted aggregate of responses after reweighting for nonresponse. District estimates (with 95% confidence intervals) were the weighted aggregate of responses after reweighting for sampling fraction and nonresponse. National estimates (with 95% confidence intervals) were the weighted aggregate of central, provincial and district level estimates after reweighting for sampling fraction (at district level) and nonresponse (at all levels). Data analysis was performed using R 4.0 (R Foundation for Statistical Computing, Vienna, Austria) [17].

## 3. Results

A total response rate of 74% (138/187 hospitals) was obtained. The response rate was 100% (6/6) for central hospitals, 72% (52/72) for provincial hospitals and 73% (80/109) for district hospitals. Table 1 displays overall and stratified estimation of KMC implementation. National estimate of KMC implementation was 49% (95% CI 40 to 57%), with decreasing estimates from 83% in central hospitals to 35% in district hospitals (Table 1).

Table 2 and Table 3 display organizational and practical aspects about KMC in hospitals where KMC is routinely performed or planned to be introduced in the next year. Overall, KMC is routinely presented to parents in 86% of the hospitals (95% CI 78 to 95%), while around half of them has a written protocol (48%, 95% CI 36 to 60%) and have held KMC courses (51%, 95% CI 39 to 63%) (Table 2). The most frequent duration of skin-to-skin contact was <6 h/day (42%, 95% CI 31 to 54%). Exclusive breastfeeding at discharge was estimated in 89% of the hospitals (95% CI 85 to 92%), while early discharge was considered in 54% (95% 42 to 65%) and follow-up to all babies who received KMC in 39% (28 to 51%) (Table 3).

Barriers to and facilitators of the implementation of KMC according to participants are summarized in Table 4. The most difficult aspects in KMC implementation were the follow-up after discharge (76%, 95% CI 68 to 84%) and achieving an adequate time spent in skin-to-skin contact (33%, 95% CI 24 to 41%). The most important facilitators were the education of parents (mean 4.2 points out of 5, 95% CI 3.9 to 4.4) and doctors (mean 3.8 points out of 5, 95% CI 3.6 to 4.0), and the availability of a KMC-dedicated room (mean 3.7 out of 5, 95% 3.5 to 3.9).

## 4. Discussion

This survey summarized the implementation of KMC in larger Vietnamese public hospitals and investigated opinions by local caregivers on barriers and strategy for adopting KMC in this setting. 

Overall, we estimated a limited implementation of KMC in larger Vietnamese maternity hospitals, with marked variations across the different levels of maternity services. As KMC is mainly recommended for preterm babies, we believe that the larger number of preterm births cared for in higher level hospitals may lead to more attention to KMC. However, barriers to and facilitators of the implementation of KMC have received a lot of interest because of its slow uptake, despite the accumulating evidence on its effectiveness for over 20 years and its promotion for over 10 years [13,18,19].

Continuous and prolonged skin-to-skin contact between mother (or family member) and newborn is the first step of KMC [1,2]. However, our findings confirmed previous reports about limited duration of skin-to-skin contact, which remains one of the main challenges for full implementation of KMC [20]. Spending an adequate time in skin-to-skin contact was also perceived as a barrier in around one third of the hospitals. We believe that the complexity of this challenge involves the availability of a suitable environment and the commitment of both parents and health staff. 

A KMC-dedicated room has been suggested as useful transition from intermittent KMC in the nursery to continuous KMC at home. In such environment, each mother can earn experience and achieve confidence before discharge, while sharing support with other mothers [20]. Our data indicated the presence of a KMC-dedicated room in about half of the hospitals, and participants recognized its importance in the implementation of KMC, thus highlighting the need for further efforts for individuating and arranging dedicated spaces.

Family support is important for the success of KMC, with the involvement of the partner and other family members playing a crucial role in helping the mother [20,21,22]. In our survey, participants acknowledged the importance of parental education to improve KMC practice. Parents frequently received oral information about KMC, while written material (brochures and posters) were less used. However, our survey showed a limited involvement of the partner and other family members, with the lowest participation in district hospitals. We believe that the small number of preterm babies in the latter may partially explain this finding. Stressing the importance of skin-to-skin contact in both hospital staff and family members, as well environmental modifications (i.e., KMC-dedicated room), can enhance the commitment of all caregivers.

In addition, we believe that the vague indication of “continuous and prolonged contact” (which hampers both the interpretation of the guidelines by the health staff and the communication between parents and health staff) may contribute to the complexity of this challenge. A previous systematic review reported significant heterogeneity in the definition of KMC in the dedicated literature [23].

Education of health care providers has a key role in the success of care implementation [20,22], as recognized by our participants. However, approximately half of the hospitals had a written protocol for KMC and hosted educational courses on KMC. Despite the KMC guidelines recommend adherence to a written protocol, participants rated this aspect as medium important. Among health care providers, doctors played a principal role in informing parents about KMC and in deciding when starting such procedure, while nurses and midwives were less frequently involved. Eligibility for KMC showed some room for improvements, especially in lower-level hospitals and in babies requiring advanced treatment (i.e., CPAP and IV fluids).

Of note, our data estimated a good prevalence of exclusive breastfeeding at discharge in babies who received KMC. As such, exclusive breastfeeding was barely perceived by our participants as barrier to KMC implementation. Exclusive breastfeeding has well-known benefits on both maternal and neonatal short- and long-term outcomes, such as delayed return to fertility, improved bonding, reduced neonatal gastro-intestinal infections and mortality, improved neurodevelopmental outcomes, and reduced chronic disease in adult age [4,24,25,26,27].

Although babies receiving KMC can be discharged early from the health facility [1,2], we estimated that around half of the hospitals do not consider early discharge for such patients. This finding coupled with the low implementation of follow-up for babies who received KMC, as these two aspects are strictly interrelated. In fact, early discharge may be offered when appropriate follow-up (from health facilities to home) can be guaranteed, thus a strong health system with motivated health care providers and adequate referral organization is warranted to overcome those obstacles [20]. Of note, less than half of the hospitals was able to provide information about breastfeeding at 6 months. Participants to our survey indicated follow-up after discharge as the most difficult aspects in KMC implementation, in agreement with previous reports [20,27]. Of note, we believe that early discharge was not perceived as a barrier because it is strongly linked with the follow-up issue, and overcoming the latter would in turn allow the early discharge of babies receiving KMC.

An expert consensus indicated the lack of KMC coverage data among the bottlenecks for KMC uptake [12]. This situation was mainly imputed to the difficulty of capturing KMC components and lack of integration of KMC data in health information systems and/or periodic surveys. Our survey found an encouraging level of recording of KMC procedures in patient’s record, but there is still room for improvement at both local and national level.

The present survey shows a slow uptake of KMC in Vietnam according to the experiences reported in other Asian countries (India, Indonesia and the Philippines) [28].

The strengths of this survey include a nationally representative sample, the large number of births at the participating hospitals, and the good response rate. Our survey provides information about adoption of KMC in Vietnam to clinicians, stakeholders and policy makers, and offers a basis for further scale up of KMC. Areas of improvements include increasing the duration of skin-to-skin contact (all levels), arranging dedicated spaces for KMC (all levels), involving the relatives (especially at district level), extending the availability of a written protocol (all levels), improving the eligibility process (all levels), and implementing early discharge and follow-up monitoring (all levels). Although these are important information, implementation of KMC at country level remains a challenge that requires operational national policies and adequate funds [28].

Nonetheless, our findings should be considered within the limitations of the study. First, survey forms were filled in by the director of the obstetrical/neonatal wards of each participating hospital and the provided information was not verified versus observed clinical practice. Second, the questionnaire was developed for this study and was not validated. Third, the three hospital levels (central, provincial, district) had different size, while we chose to weight our results by hospital level rather than number of births. However, we believed this better represented the policy task, by focusing on the proportion of hospitals requiring supplementary policy support and training to enhance the implementation of KMC [14,15]. Although the statistical analysis contains a substantial correction we believe that it well represents all the Vietnamese hospital levels as previously reported [14,15].

## 5. Conclusions

We estimated a limited implementation of KMC in larger Vietnamese maternity hospitals, with marked variations across the different levels of maternity services. We also reported insights on barriers to and facilitators of the implementation of KMC according to local health care providers. This survey indicates strengths and limitations in KMC implementation in larger Vietnamese hospitals, suggesting possible areas of improvements to clinicians, stakeholders and policy makers for further scale up of KMC.

## Figures and Tables

**Table 1 children-09-01667-t001:** Implementation of Kangaroo Mother Care (in all participating hospitals).

		Hospital Level		
	Overall Estimate: % (95% CI)	Central: %	Provincial: % (95% CI)	District: % (95% CI)
**Routine KMC implementation**:				
KMC is routinely performed	49% (40 to 57%)	83%	65% (59 to 72%)	35% (25 to 45%)
Planning to introduce KMC in the next 12 months	14% (9 to 20%)	17%	19% (14 to 25%)	11% (5 to 18%)
KMC is not routinely performed	37% (29 to 45%)	0%	16% (20 to 21%)	54% (44 to 64%)

Confidence intervals (CI) were calculated for overall, provincial and district estimates, while estimates at central level were the percentages calculated among responders, as the study included the census of all central hospitals (with 100% response rate).

**Table 2 children-09-01667-t002:** Organizational aspects of Kangaroo Mother Care (in hospitals where KMC is routinely performed or planned to be introduced in the next 12 months).

		Hospital Level		
	Overall Estimate: % (95% CI)	Central: %	Provincial: % (95% CI)	District: % (95% CI)
**Availability of written protocol for KMC**	48% (36 to 60%)	67%	55% (47 to 62%)	46% (31 to 61%)
**Starting KMC is decided by: ^a^**				
Doctor	95% (90 to 100%)	100%	98% (95 to 100%)	94% (88 to 100%)
Nurse	23% (13 to 33%)	17%	27% (20 to 34%)	22% (9 to 34%)
Midwife	41% (29 to 52%)	0%	23% (16 to 29%)	46% (31 to 61%)
**Infants usually considered eligible for KMC: ^a^**				
Full-term infants	47% (35 to 59%)	50%	27% (20 to 34%)	51% (36 to 66%)
Late preterm infants (34–36 weeks)	84% (76 to 92%)	67%	77% (71 to 84%)	86% (76 to 97%)
Very preterm infants (≤33 weeks)	40% (29 to 51%)	100%	64% (56 to 71%)	32% (18 to 46%)
Babies receiving CPAP	14% (6 to 22%)	33%	14% (8 to 19%)	14% (3 to 24%)
Babies receiving continuous IV fluids	19% (10 to 29%)	33%	20% (14 to 27%)	19% (7 to 31%)
Babies with very LBW <1.5 kg	50% (38 to 62%)	83%	64% (56 to 71%)	46% (31 to 61%)
Babies with extremely LBW <1.2 kg	34% (23 to 45%)	83%	57% (49 to 65%)	27% (14 to 40%)
**Educational courses on KMC have been held in the hospital ^b^**	51% (39 to 63%)	50%	59% (51 to 67%)	49% (34 to 64%)
**Posters to advertise KMC in Vietnamese:**				
Only in postnatal ward	28% (17 to 40%)	17%	14% (8 to 19%)	32% (18 to 46%)
Only in NICU	7% (1 to 12%)	0%	14% (8 to 19%)	5% (0 to 12%)
In both postnatal ward and NICU	17% (9 to 26%)	33%	32% (24 to 39%)	14% (3 to 24%)
No posters available in the hospital	47% (35 to 59%)	50%	41% (33 to 49%)	49% (33 to 64%)
**Routine presentation of KMC to parents** ^c^	86% (78 to 95%)	83%	86% (81 to 92%)	86% (76 to 97%)

Confidence intervals (CI) were calculated for overall, provincial and district estimates, while estimates at central level were the percentages calculated among responders, as the study included the census of all central hospitals (with 100% response rate). CPAP: continuous positive airway pressure. KMC: Kangaroo Mother Care. LBW: low birth weight. NICU: neonatal intensive care unit. ^a^ Hospitals could indicate more than one aspect. ^b^ Educational courses: (i) were held at median once per year, (ii) were mostly based on Essential Newborn Care-World Health Organization guidelines (77%) and/or governmental guidelines (30%), (iii) the majority have a duration of one (28%) or three (40%) days, and (iv) the trainers were mainly local (in-hospital) instructors (47%) or governmental instructors (40%). ^c^ Parents received information on KMC by doctors (80%), nurses (61%) and/or midwives (50%), with sometimes provision of written material (42%).

**Table 3 children-09-01667-t003:** Practical aspects of Kangaroo Mother Care (in hospitals where KMC is routinely performed or planned to be introduced in the next 12 months).

		Hospital Level		
	Overall Estimate: % (95% CI)	Central: %	Provincial: % (95% CI)	District: % (95% CI)
**KMC is performed by: ^a^**				
Only mother	53% (42 to 65%)	20%	25% (18 to 32%)	61% (46 to 75%)
Mother and father	32% (21 to 42%)	60%	50% (42 to 58%)	26% (13 to 39%)
Only father	3% (0 to 7%)	20%	5% (1 to 8%)	3% (0 to 7%)
Other family members (when parents are unavailable)	32% (22 to 42%)	80%	61% (54 to 69%)	24% (11 to 36%)
**KMC is performed in: ^a^**				
KMC-dedicated room ^b^	45% (33 to 56%)	60%	66% (58 to 73%)	39% (25 to 54%)
Mother’s room	40% (28 to 51%)	40%	30% (22 to 37%)	42% (28 to 57%)
Neonatal intensive care unit	15% (7 to 23%)	80%	16% (10 to 22%)	13% (3 to 23%)
Other rooms	28% (17 to 39%)	0%	14% (8 to 19%)	32% (18 to 45%)
**Temperature control in the room where KMC is performed:**				
Climate-controlled room ^c^	67% (55 to 78%)	100%	79% (73 to 86%)	63% (49 to 77%)
Depending on the season (not climate controlled)	26% (15 to 37%)	0%	16% (10 to 22%)	29% (16 to 42%)
Unknown	7% (0 to 13%)	0%	5% (1 to 8%)	8% (0 to 16%)
**Average daily duration of SSC during KMC:**				
<6 h per day (0–25% of day)	42% (31 to 54%)	0%	25% (18 to 32%)	46% (33 to 62%)
6–12 h per day (25–50% of day)	13% (5 to 21%)	40%	11% (6 to 16%)	13% (3 to 23%)
12–18 h per day (50–75% of day)	25% (15 to 25%)	20%	30% (22 to 37%)	24% (11 to 36%)
18–24 h per day (75–100% of day)	9% (5 to 13%)	40%	32% (23 to 39%)	3% (0 to 7%)
Unknown	11% (3 to 19%)	0%	2% (0 to 5%)	13% (3 to 23%)
**Duration of SSC during KMC is decided by:**				
The mother	15% (7 to 24%)	0%	14% (8 to 19%)	16% (5 to 27%)
Both the mother and the health care staff	1% (0 to 2%)	0%	5% (1 to 8%)	NA
The health care staff	53% (41 to 64%)	40%	32% (24 to 39%)	58% (43 to 72%)
Specific duration based on our local protocol	23% (14 to 32%)	60%	50% (42 to 58%)	16% (5 to 27%)
Unknown duration	8% (1 to 16%)	0%	NA	11% (1 to 20%)
**Measurement of baby’s body temperature during KMC**	58% (47 to 70%)	80%	82% (76 to 88%)	53% (38 to 67%)
**Number of measurements per day:**				
Pre-established number ^d^	57% (43 to 72%)	50%	78% (71 to 85%)	50% (30 to 70%)
Based on health staff decision	34% (20 to 48%)	50%	17% (10 to 23%)	40% (20 to 60%)
Unknown	9% (0 to 17%)	0%	6% (2 to 10%)	10% (0 o 22%)
**Exclusive breastfeeding at discharge**	89% (85 to 92%)	80%	84% (78 to 90%)	92% (84 to 100%)
**Early discharge for babies who received KMC:**				
Yes	54% (42 to 65%)	60%	68% (61 to 76%)	50% (35 to 65%)
No	36 (24 to 47%)	40%	30% (22 to 37%)	37% (23 to 51%)
Unknown	11% (3 to 19%)	0%	2% (0 to 5%)	1% (3 to 23%)
**Follow-up for babies who received KMC:**				
All babies who received KMC	39% (28 to 51%)	20%	30% (22 to 37%)	42% (28 to 57%)
Only preterm babies who received KMC	23% (15 to 32%)	60%	52% (44 to 60%)	16% (5 to 27%)
Sickest babies who received KMC	1% (0 to 2%)	20%	5% (1 to 8%)	NA
None	29% (18 to 40%)	0%	9% (5 to 14%)	34% (20 to 48%)
Unspecified	7% (1 to 13%)	0%	5% (1 to 8%)	8% (0 to 16%)
**Availability of information about breastfeeding at 6 months**	44% (32 to 56%)	40%	32% (24 to 39%)	47% (33 to 62%)
**Documentation on KMC in patient’s record:**				
Yes	71% (60 to 81%)	80%	68% (61 to 76%)	71% (58 to 84%)
No	23% (13 to 33%)	20%	32% (24 to 39%)	21% (9 to 33%)
Unspecified	6% (0 to 13%)	0%	NA	8% (0 to 16%)

Confidence intervals (CI) were calculated for overall, provincial and district estimates, while estimates at central level were the percentages calculated among responders, as the study included the census of all central hospitals (with 100% response rate). KMC: Kangaroo Mother Care. NA: not available. SSC: skin-to-skin contact. ^a^ Hospitals could indicate more than one aspect. ^b^ The KMC-dedicated room had a median of 4 beds/chairs. ^c^ In climate-controlled rooms where KMC was performed, median temperature was 28.8 °C. ^d^ Median twice per day.

**Table 4 children-09-01667-t004:** Barriers to and facilitators of the implementation of Kangaroo Mother Care.

		Hospital Level		
Most Difficult Aspects in KMC Implementation: ^a^	Overall Estimate: % (95% CI)	Central: %	Provincial: % (95% CI)	District: % (95% CI)
Adequate time spent in skin-to-skin contact	33% (24 to 41%)	33%	42% (35 to 49%)	31% (22 to 41%)
Exclusive breastfeeding	14% (8 to 20%)	17%	23% (17 to 29%)	13% (6 to 19%)
Early discharge (whenever possible)	7% (2 to 12%)	0%	4% (1 to 7%)	8% (2 to 13%)
Follow-up after discharge	76% (68 to 83%)	50%	83% (77 to 88%)	75% (66 to 84%)
**Importance (from 1 to 5) of strategies to implement/improve KMC:**	**Overall estimate:** **mean (95% CI)**	**Central:** **mean**	**Provincial:** **mean (95% CI)**	**District:** **mean (95% CI)**
Education of the mother	3.3 (3.1 to 3.4)	2.8	3.0 (2.9 to 3.1)	3.3 (3.1 to 3.5)
Education of mother and father	4.2 (3.9 to 4.4)	4.8	4.3 (4.1 to 4.4)	4.1 (3.9 to 4.4)
Education of doctors	3.8 (3.6 to 4.0)	3.7	3.9 (3.7 to 4.0)	3.8 (3.6 to 4.1)
Education of nurses	3.3 (3.1 to 3.6)	4.3	3.9 (3.7 to 4.1)	3.3 (3.0 to 3.5)
Education of midwifes	3.5 (3.3 to 3.7)	2.7	3.0 (2.7 to 3.2)	3.5 (3.3 to 3.8)
Availability of a KMC-dedicated room	3.7 (3.5 to 3.9)	4.7	3.8 (3.6 to 3.9)	3.7 (3.4 to 4.0)
Close follow-up after discharge	2.6 (2.4 to 2.9)	3.5	2.7 (2.5 to 2.9)	2.6 (2.3 to 2.9)
Improve babies/nurse or babies/midwife ratio	2.3 (2.1 to 2.5)	3	2.2 (2.0 to 2.3)	2.3 (1.2 to 2.5)
Written protocol on KMC	3.0 (2.8 to 3.2)	3	3.0 (2.8 to 3.2)	3.0 (2.7 to 3.2)
Lack of other equipment	3.3 (3.1 to 3.5)	3.7	3.2 (3.0 to 3.4)	3.3 (3.0 to 3.5)
Lack of temperature control during summer	3.1 (2.9 to 3.3)	3.2	3.2 (3.0 to 3.3)	3.1 (2.9 to 3.3)

Confidence intervals (CI) were calculated for overall, provincial and district estimates, while estimates at central level were the percentages calculated among responders, as the study included the census of all central hospitals (with 100% response rate). KMC: Kangaroo Mother Care. ^a^ Hospitals could indicate more than one aspect.

## Data Availability

Original data are available upon reasonable request to the corresponding author.

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
