# Peer review of "Kangaroo Mother Care in Vietnam: A National Survey of a Middle-Income Country"

_children, 2022, doi:10.3390/children9111667_

Round 1

Reviewer 1 Report

Thank you very much to allow me to review the article entitle “Kangaroo Mother Care in Vietnam: a national survey of a middle-income country” (children-1973010), which is presented to the Section “Global and Public Health”.

The aim of this work was investigated the current provision of Kangaroo mother care in Vietnam and explored differences among levels of healthcare facility.

Introduction : Use adequate and relevant bibliography to provide information about what Kangaroo mother care (KMC) is. I suggest indicating how long this program has been running in Vietnam in order to be able to assess the status of the program taking into account the time it has been in place. As well as what proportion of the population is treated in public hospitals where the program is carried out

The objective is clearly presented.

Material and methods: Although data from a survey of hospitals are used, as it is an investigation, it is recommended that the approval of the ethics committee that has allowed the conduct of this study be presented.

The design used should be indicated.

Given that there are 532 hospital districts and that finally 187 hospitals participate, it should be explained whether the sample of central, provincial and hospital district hospitals is proportional or not to these centers in Vietnam.

Is the survey used validated, or is it an add hoc survey?

Is there a difference between premature and low birth weight?

Results:

The tables presented in results are very well designed and present the results clearly, allowing a better understanding of the results obtained.

Great information is presented with great precision. However, I suggest that it would be appropriate to attach a text explaining the most outstanding results to further facilitate the understanding of the results.

The discussion is undoubtedly very interesting and complete, but I suggest adding what is detected as improvable at each level, as well as proposals for improvement based on the scientific evidence presented.

Author Response

1.Introduction: Use adequate and relevant bibliography to provide information about what Kangaroo mother care (KMC) is. I suggest indicating how long this program has been running in Vietnam in order to be able to assess the status of the program taking into account the time it has been in place. As well as what proportion of the population is treated in public hospitals where the program is carried out.

Re: Unfortunately, data on the roll out of the KMC program in Vietnam are unclear. In 2014, the Vietnam Ministry of Health endorsed a newborn program (Action Plan for Healthy Newborn Infants in the Western Pacific Region 2014–2020) including early essential newborn care and KMC, but some data indicated that KMC was already adopted in some hospitals. We included the proportion of the population in the revised manuscript: “The sampling frame for the study included the 610 maternity hospitals with 500 or more births per year, representing 97% of maternity hospital births [14].” (page 2)

2.The objective is clearly presented.

Re: We thank the Reviewer for the comment.

3.Material and methods: Although data from a survey of hospitals are used, as it is an investigation, it is recommended that the approval of the ethics committee that has allowed the conduct of this study be presented.

Re: To our knowledge, ethics approval is not required for this type of study, as our survey does not include human or clinical data and/or patients’ information but only information on KMC practices and barriers/facilitators in the participating hospitals.

4.The design used should be indicated.

Re: The design (cross-sectional) was included in Methods section: “A structured, cross-sectional survey …..” (page 2).

5.Given that there are 532 hospital districts and that finally 187 hospitals participate, it should be explained whether the sample of central, provincial and hospital district hospitals is proportional or not to these centers in Vietnam.

Re: We specified these aspects in the Methods section: “The sampling frame for the study included the 610 maternity hospitals with 500 or more births per year, representing 97% of maternity hospital births [14]. This list was used to create two samples: the first included the census of all 6 central hospitals and 72 provincial hospitals, while the second included a 20 % sample survey of 532 district hospitals. These were randomly selected from each Vietnamese administrative region: each region provided 20% of the district hospitals in its catchment, using a stratified random sampling with proportional allocation approach [14,15]. Overall, 187 hospitals (6 central hospitals, 72 provincial hospitals, 109 district hospitals) were contacted to participate in the survey.” (page 2), and “As the response rate was <100% at the provincial and district levels, data were inflated by the inverse of the response rate to avoid systematic bias.” (page 3).

6.Is the survey used validated, or is it an ad-hoc survey?

Re: The survey was ad-hoc, hence we added this among the limitations of the study: “Second, the questionnaire was developed for this study and was not validated.“ (Discussion, page 7).

7.Is there a difference between premature and low birth weight?

Re: We found that late preterm infants (34-36 weeks) were considered eligible for KMC more frequently than very preterm infants (33 weeks or less) and very low birth weight infants (<1.5 kg), as shown by the not-overlapping confidence intervals in Table 2.

  1. Results: The tables presented in results are very well designed and present the results clearly, allowing a better understanding of the results obtained. Great information is presented with great precision. However, I suggest that it would be appropriate to attach a text explaining the most outstanding results to further facilitate the understanding of the results.

Re: We thank the Reviewer for the suggestion. We added some lines highlighting the main findings, while avoiding the duplication of the information in the tables: “A total response rate of 74% (138/187 hospitals) was obtained. The response rate was 100% (6/6) for central hospitals, 72% (52/72) for provincial hospitals and 73% (80/109) for district hospitals. Table 1 displays overall and stratified estimation of KMC implementation. National estimate of KMC implementation was 49% (95% CI 40 to 57%), with de-creasing estimates from 83% in central hospitals to 35% in district hospitals (Table 1). Tables 2 and 3 display organizational and practical aspects about KMC in hospitals where KMC is routinely performed or planned to be introduced in the next year. Overall, KMC is routinely presented to parents in 86% of the hospitals (95% CI 78 to 95%), while around half of them has a written protocol (48%, 95% CI 36 to 60%) and have held KMC courses (51%, 95% CI 39 to 63%) (Table 2). The most frequent duration of skin-to-skin contact was <6 hours/day (42%, 95% CI 31 to 54%). Exclusive breastfeeding at discharge was estimated in 89% of the hospitals (95% CI 85 to 92%), while early discharge was considered in 54% (95% 42 to 65%) and follow-up to all babies who received KMC in 39% (28 to 51%) (Table 3). Barriers to and facilitators of the implementation of KMC according to participants are summarized in Table 4. The most difficult aspects in KMC implementation were the follow-up after discharge (76%, 95% CI 68 to 84%) and achieving an adequate time spent in skin-to-skin contact (33%, 95% CI 24 to 41%). The most important facilitators were the education of parents (mean 4.2 points out of 5, 95% CI 3.9 to 4.4) and doctors (mean 3.8 points out of 5, 95% CI 3.6 to 4.0), and the availability of a KMC-dedicated room (mean 3.7 out of 5, 95% 3.5 to 3.9).” (pages 3-6).

9.The discussion is undoubtedly very interesting and complete, but I suggest adding what is detected as improvable at each level, as well as proposals for improvement based on the scientific evidence presented.

Re: We included these considerations: “Areas of improvements include increasing the duration of skin-to-skin contact (all levels), arranging dedicated spaces for KMC (all levels), involving the relatives (especially at district level), extending the availability of a written protocol (all levels), improving the eligibility process (all levels), and implementing early discharge and follow-up monitoring (all levels).” (Discussion, pages 7-8).

Reviewer 2 Report

The study entitled “Kangaroo Mother care in Vietnam: a national survey of a middle-income country” shows a detailed description of the implementation of the KMC in this country as well as the analysis of organizational aspects at the different levels of medical assistance.

The manuscript is certainly well written, and the findings offer an accurate picture of the KMC practice at Vietnamese hospitals.

There are some aspects of this text that could be clarified or corrected:

1.      Abstract: The conclusions don’t match with the results that have been included. I would replace some sentences regarding the implementation of KMC by more specific results such as the included in table 1.

2.      The statistical analysis contains a substantial correction introduced by the authors that should be pointed out and justified at the discussion.

3.       The discussion is limited to a summary of the results. It would be desirable to find comparisons with other countries or the impact of KMC in this area, considering socioeconomic and/or health consequences.

4.      Line 232: the word “manly” probably meant “mainly”.

Author Response

The study entitled “Kangaroo Mother care in Vietnam: a national survey of a middle-income country” shows a detailed description of the implementation of the KMC in this country as well as the analysis of organizational aspects at the different levels of medical assistance.

The manuscript is certainly well written, and the findings offer an accurate picture of the KMC practice at Vietnamese hospitals.

There are some aspects of this text that could be clarified or corrected:

  1. Abstract: The conclusions don’t match with the results that have been included. I would replace some sentences regarding the implementation of KMC by more specific results such as the included in table 1.

Re: We have changed the coonclusion section adding the following part: “Areas of improvements include increasing the duration of skin-to-skin contact, arranging dedi-cated spaces for KMC, involving the relatives (especially at district level), extending the availa-bility of a written protocol, improving the eligibility process, and implementing early discharge and follow-up monitoring.” (Page 1)

  1. The statistical analysis contains a substantial correction introduced by the authors that should be pointed out and justified at the discussion.

Re: We added the following sentence among the limitations of the study: “Although the statistical analysis contains a substantial correction we believe that it well represents all the Vietnamese hospital levels as previously reported [14,15].” (Page 8)

  1. The discussion is limited to a summary of the results. It would be desirable to find comparisons with other countries or the impact of KMC in this area, considering socioeconomic and/or health consequences.

Re: We thank the reviewer for this suggestion. We added the following parts:

“The present survey shows a slow uptake of KMC in Vietnam according to the experi-ences reported in other Asian countries (India, Indonesia and the Philippines) [29].” (Page 7)

“). Although these are important information, implementation of KMC at country level re-mains a challenge that requires operational policies and adequate funds included in the national health  plans [29].” (page 8)  

Ref#29.

  1. Line 232: the word “manly” probably meant “mainly”.

Re: Corrected.